# Effects of Marinades Prepared from Food Industry By-Products on Quality and Biosafety Parameters of Lamb Meat

**DOI:** 10.3390/foods12071391

**Published:** 2023-03-24

**Authors:** Paulina Zavistanaviciute, Jolita Klementaviciute, Dovile Klupsaite, Egle Zokaityte, Modestas Ruzauskas, Vilija Buckiuniene, Pranas Viskelis, Elena Bartkiene

**Affiliations:** 1Institute of Animal Rearing Technologies, Faculty of Animal Sciences, Lithuanian University of Health Sciences, Tilzes Str. 18, LT-47181 Kaunas, Lithuania; 2Department of Food Safety and Quality, Veterinary Academy, Lithuanian University of Health Sciences, Tilzes Str. 18, LT-47181 Kaunas, Lithuania; 3Department of Anatomy and Physiology, Faculty of Veterinary, Lithuanian University of Health Sciences, Tilzes Str. 18, LT-47181 Kaunas, Lithuania; 4Institute of Microbiology and Virology, Faculty of Veterinary, Lithuanian University of Health Sciences, Mickeviciaus Str. 9, LT-44307 Kaunas, Lithuania; 5Lithuanian Research Centre for Agriculture and Forestry, Institute of Horticulture, Kauno Str. 30, LT-54333 Babtai, Lithuania

**Keywords:** acid whey, apple by-products, blackcurrant by-products, lamb meat, marinades, quality parameters, safety parameters, lactic acid bacteria, valorisation

## Abstract

This study aimed to develop marinade formulas based on by-products from the dairy, berry, and fruit industries and apply them to lamb meat (LM) treatments to improve the safety and quality characteristics of the meat. To fulfil this aim, six marinade (M) formulations were created based on acid whey (AW) fermented with *Lacticaseibacillus casei* (Lc) and *Liquorilactobacillus uvarum* (Lu), either alone or combined with freeze-dried apple (AP) or blackcurrant (BC) pomace. The most appropriate fermentation times for the marinades were selected according to the lower pH values and higher viable LAB counts in the samples. Additionally, the antimicrobial activity of the selected marinades against pathogenic and opportunistic bacterial strains was tested. The characteristics of the LM were analysed after 24 and 48 h of treatment, including physicochemical, technological, and microbiological parameters, as well as overall acceptability. It was established that, after 48 h of fermentation, all of the tested marinades, except M-AW_Lu_BC, had lactic acid bacterial counts > 8.0 log_10_ CFU·mL^−1^ and pH values < 3.74. The broadest spectra of pathogen inhibition were observed in the M-AW_Lu_AP and M-AW_Lu_BC marinades. The latter formulations improved the water holding capacity (WHC) and overall acceptability of the LM, while, in the LM-AW_Lc_AP samples, histamine, cadaverine, putrescine, tryptamine, and phenylethylamine were not formed. Lastly, LM treatment with the M-AW_Lc_AP and M-AW_Lu_AP formulas for 48 h achieved the highest overall acceptability (9.04 and 9.43), tenderness (1.53 and 1.47 kg·cm^−2^) and WHC (2.95% and 3.5%) compared to the control samples.

## 1. Introduction

Meat plays an essential role in our food culture, especially in the Western world, and it is a vital source of proteins in human nutrition [1,2]. However, about 23% of production in the meat sector, including all stages of the food chain, is generated as waste [3]. The majority of losses and waste in Europe occur at the retail and consumer levels, i.e., at the very end of the food supply chain [4,5]. The main aspects causing food losses are unappropriated storage conditions, expired shelf lives, shelf lives that are too short, and mechanical damage to packaging, which lead to the deterioration of the quality of the products [3,4,6]. In Europe, the generation of lamb meat (LM) waste is about 5% [6]; however, information about the amount of LM waste worldwide is limited. The intake of lamb meat (LM) is growing, and the socioeconomic, ethical, religious and psychological characteristics of consumers play important roles in the rising consumption levels [7,8]. LM contains essential amino and fatty acids, as well as micronutrients (e.g., Fe, Zn, and Se, as well as vitamins B3 and B12 and folic acid) [2,9,10], which play important roles in human nutrition [7,11]. However, LM consumption is restricted by its specific odour, taste, texture, and short shelf life [7,12,13,14]. Different technologies have been applied to improve LM quality parameters, e.g., sonication [15,16] and thermal processing [7,17,18]. Additionally, the pre-treatment of LM with natural marinades has also been reported [19,20,21]. This solution is of great interest because, currently, consumers are looking for natural, high-quality, and acceptable products. Additionally, food industry by-products can be used for marinade preparation as they are valuable sources of bioactive compounds, including antimicrobial compounds. Furthermore, the addition of viable technological lactic acid bacteria (LAB) strains could be a prospective solution for the preparation of marinades.

Synthetic food additives can improve the colour and flavour and extend the shelf lives of products [22]. For instance, sorbic and benzoic acids, as well as their salts, produce mutagenic and carcinogenic compounds [23,24]. However, nitrites and nitrates, which affect the colour of meat and its products, are related to leukaemia, colon cancer, bladder cancer, and others health issues [22,25,26]. Due to the adverse effects of synthetic food additives on consumer health, there is increasing demand for natural substances to be used as food additives, especially meat treatments [22,27,28]. It has been found that compounds from spices, herbs, or tea can improve the quality parameters of LM; however, their extraction requires expensive technologies [28,29,30]. LAB are generally recognised as safe and offer plenty of advantages: (I) they can improve flavour and texture [19,31]; (II) due to their antimicrobial [32,33] and antifungal [33,34] metabolites, they can be used as biopreservatives to extend shelf lives [34,35]; (III) they have antioxidant properties [36,37]. It has also been published that lactobacilli strains have a broad range of applications, including meat processing [19,31,38,39]. As commercial media for LAB propagation are expensive, alternative and environmentally friendly solutions should be explored. One alternative is enriched acid whey (AW) [40]. Considering that AW production has increased because of the rising demand for Greek yoghurt and acid-coagulated cheeses [41], now would be a good time to valorise it. According to Augustyńska-Prejsnar et al. [42], AW can be used as an organic marinade for chicken meat treatment. It should be mentioned that the sensory properties of AW are not very acceptable to consumers, but fermented AW can be enriched with berry or fruit by-products to improve its sensory characteristics. The application of berry/fruit by-products is limited due to the high amount of water that is left in the pomace [32]. However, various valorisation techniques (e.g., freeze-drying) can not only remove moisture but also preserve functional components [43,44], such as phenolic compounds (i.e., phenolic acids, flavonoids, such as anthocyanins and flavonols, and tannins) and ascorbic acid [44,45]. Our previous studies have shown that freeze-dried apple pomace (AP) and freeze-dried blackcurrant pomace (BC) possess high antimicrobial [32,36,37,46] and antifungal [40,47] activity against a broad spectrum of pathogenic bacteria strains and fungi. It is also known that by-products from the berry/fruit industry can improve the sensory properties (i.e., colour, flavour, moisture, and texture) of different types of foods [36,46,48,49,50,51]. The influence of juices, extracts and vinegar has also been reported on the quality and sensory parameters of pork [52,53,54], poultry [53,54,55,56], beef [57,58,59,60], and LM [61,62]; however, information about marinades prepared using AW–LAB–berry/fruit combinations is scarce. 

In this study, we hypothesised that sustainable marinade formulas developed from dairy (fermented AW), berry, and fruit industry by-products could lead to the higher quality and improved biosafety of LM preparation.

The aim of this study was to develop marinade formulas based on dairy, berry, and fruit industry by-products and apply them in LM treatments to improve the safety and quality characteristics of the LM. To fulfil this aim, six marinade formulations were developed, based on acid whey (AW): (I) AW fermented with *Lacticaseibacillus casei* LUHS210 (Lc); (II) AW fermented with *Liquorilactobacillus uvarum* LUHS245 (Lu); AW fermented with Lc in combination with freeze-dried (III) apple (AP) and (IV) blackcurrant (BC) pomace; AW fermented with Lu in combination with freeze-dried (V) apple (AP) and (VI) blackcurrant (BC) pomace. The most appropriate marinade formulas were selected according to their pH values and viable LAB counts. Additionally, the antimicrobial activity of the selected marinades was tested against *Salmonella enterica*, *Pseudomonas aeruginosa*, *Proteus mirabilis*, *Enterococcus faecalis*, *Enterococcus faecium*, *Bacillus cereus*, *Streptococcus mutans*, and *Citrobacter freundii*. The following characteristics of the LM were analysed after 24 and 48 h of treatment: physicochemical, technological and microbiological parameters, and overall acceptability.

## 2. Materials and Methods

### 2.1. Ingredients Used for Experiments 

The lamb (age: 6 months; sex: male; breed: Suffolk) meat (*Musculus gluteus*) used for the experiments was purchased from a local market (Kaunas, Lithuania). The LM treatments with the prepared marinades were performed at 48 h post mortem.

The freeze-dried apple (variety: Auksis) pomace (AP) and blackcurrant (variety: Ben Alder) pomace (BC) were obtained from the Institute of Horticulture, Lithuanian Research Centre for Agriculture and Forestry (Babtai, Kaunas District, Lithuania) in 2021. The AP and BC (consisting of peel, pulp and seeds) pomace were formed during juicing. AP and BC by-products were selected for marinade preparation according to their antimicrobial properties, as described by Bartkiene et al. [32].

For AW fermentation, *Lacticaseibacillus casei* LUH210 (Lc) and *Liquorilactobacillus uvarum* LUHS245 (Lu) were used, which were previously isolated from spontaneously fermented rye sourdough [33]. These LAB strains showed good carbohydrate metabolism, viability and stability characteristics at low pH values, and antimicrobial properties against Gram-positive and Gram-negative pathogenic bacteria strains [32,33,63]. Prior to use, the LAB strains were kept at −80 °C (PRO-LAB Diagnostics, Bromborough, UK) and supplemented with 20% glycerol (Sigma-Aldrich, Taufkirchen, Germany). For the experiments, the LAB were multiplied in MRS broth with Tween-80 (Biolife, Milano, Italy) at 30 ± 2 °C for 48 h and then used for AW fermentation and marinade preparation.

The AW (lactose: 4.10%; protein: 0.75%; lactic acid: 0.64%; fats: 0.003%; ash: 0.73%; total solids: 6.10%; pH: 3.9) was obtained from JSC Pieno zvaigzdes (Kaunas, Lithuania). 

The pathogenic and opportunistic strains (Salmonella enterica, Pseudomonas aeruginosa, Proteus mirabilis, Enterococcus faecalis, Enterococcus faecium, Bacillus cereus, Streptococcus mutans, and Citrobacter freundii) were obtained from the Lithuanian University of Health Sciences (Kaunas, Lithuania) and were used for the evaluation of the antimicrobial properties of the developed marinades. 

### 2.2. Principal Scheme of the Experiments

The principal scheme of the experiments is shown in Appendix A. In the first step of the experiments, six different formulations of marinades were developed: AW fermented with Lc (M-AW_Lc_); AW fermented with Lc in combination with freeze-dried AP pomace (M-AW_Lc_AP); AW fermented with Lc in combination with freeze-dried BC pomace (M-AW_Lc_BC); AW fermented with Lu (M-AW_Lu_); AW fermented with Lu in combination with freeze-dried AP pomace (M-AW_Lu_AP); AW fermented with Lu in combination with freeze-dried BC pomace (M-AW_Lu_BC). The details of the marinade compositions are presented in Table 1. The marinades were fermented at 32 ± 2 °C, and their characteristics were evaluated after 24 h and 48 h [49]. The best marinade formulas were selected according to their pH values and viable LAB counts. Additionally, the antimicrobial activity of the selected marinades was tested against the pathogens mentioned in Section 2.1.

During the second stage of the experiments (Appendix A), the LM was treated with the prepared marinades (Table 2). The samples were divided into seven groups: the control group (without treatment, LM-C) and six groups of LM treated with AW_Lc_, AW_Lc_AP, AW_Lc_BC, AW_Lu_, AW_Lu_AP, and AW_Lu_BC. For meat preparation, the static soak method was applied. Each sample of LM was inserted into a glass jar, covered with one of the prepared marinades, and stored in a refrigerator (+4 ± 1 °C). The LM was analysed after 24 h and 48 h of marination. The microbial, physicochemical, and technological parameters, as well as the overall acceptability, of the LM were evaluated (Appendix A).

### 2.3. Analysis of the Acidity Parameters of the Prepared Marinades

The pH values were measured using a pH electrode (PP-15; Sartorius, Goettingen, Germany). Before the experiments, the pH meter was calibrated using a pH 10.01 buffer, pH 7.00 buffer, and pH 4.01 buffer at 25 °C. Firstly, the pH electrode was rinsed with deionised water and placed into the pH 10.01 buffer for at least 1–2 min until it read a stable value. The same procedure was then repeated with the pH 7.00 buffer and pH 4.01 buffer. The temperature of the buffers was 25 °C.

The total titratable acidity (TA) was evaluated using 10 mL of the samples mixed with 90 mL of distilled water. This TA evolution method was described in detail by Bartkiene et al. [64].

### 2.4. Analysis of the Microbiological Parameters of the Marinades and the Treated Lamb Meat 

The microbiological parameters (i.e., total bacteria count (TBC), lactic acid bacteria count (LABC), total enterobacteria count (TEBC), and mould/yeast count (M/Y)) of the prepared marinades and the treated LM were evaluated. For this evaluation, 10 mL and 10 g of a sample were homogenised with 90 mL of saline (0.9%). Then, serial dilutions of 10^1^–10^7^ with saline were used for the sample preparation.

The LAB counts were determined on MRS agar with Tween-80 (Biolife, Milano, Italy) using standard plate count techniques. The procedures were described in detail in ISO 15214:1998 [65].

The total bacteria counts (TBCs) were determined on plate count agar (Biolife, Milan, Italy), as described in ISO 4833-2 [66].

The total enterobacteria counts (TEBCs) were determined on violet red bile glucose agar (Oxoid Ltd., Basingstoke, UK), as described in ISO 21528-2 [67].

The yeast and mould counts were determined on Dichloran rose Bengal chloramphenicol agar (Liofilchem, Milan, Italy), as described in ISO 21527-2 [68]. 

The total bacteria counts were calculated and expressed as the log_10_ of colony-forming units (CFU·mL^−1^ and g) of the samples. 

### 2.5. Analysis of the Antimicrobial Activity of the Prepared Marinades

The antimicrobial activity of the prepared marinades was tested against selected pathogenic and opportunistic bacterial strains (as described in Section 2.1) using an agar well diffusion assay. 

For this analysis, a 0.5 McFarland unit density suspension (~10^8^ CFU mL^−1^) of each pathogenic bacterial strain (all strains used were described in Section 2.1) was inoculated onto the surface of cooled Mueller–Hinton agar (Oxoid, Basingstoke, UK) using sterile cotton swabs [32]. Then, 50 mL of the prepared marinades were poured into wells in the agar, which had diameters of 6 mm. The results were evaluated by measuring the diameters of inhibition zones (DIZs, mm). The method was described in detail by Bartkiene et al. [32]. 

### 2.6. Analysis of the Technological and Physicochemical Parameters of the LM

The dry matter (DM) was evaluated using the method described in ISO 1442:1997 [69]. 

The LM pH values were evaluated after 24 and 48 h of treatment using a pH meter (model Inolab 3; Hanna Instruments, Remetea Mare, Romania). 

The protein contents in the LM samples were analysed using the Kjeldahl method, with a coefficient of 6.25. 

The intramuscular fat contents were determined using a Soxhlet SE 416 Columbus macro automatic system for fat extraction (Gerhardt, Königswinter, Germany). The sample preparation and method were described in detail by Klupsaite et al. [13].

The ash content was determined by burning the LM samples in a muffle furnace at 700 °C for 4 h.

The water holding capacity (WHC) was determined using a sample of approximately 2.0 g from each *Gluteus medius* muscle and the procedure was performed in triplicate. The results were expressed as the percentage of water exuded relative to the weight of the initial sample. 

Drip loss (DL) was measured as the weight loss during the suspension of a standardised muscle sample (40–50 g and approximately 30 × 60 × 25 mm in size) in an airtight container over 24 h at 4 °C [13].

To determine the cooking loss (CL) and shear force, which demonstrated the tenderness of the LM, the samples were boiled in a water bath inside a heat-resistant plastic container until reaching an internal temperature of 70 °C for 30 min and were subsequently cooled. The CL corresponded to the weight difference between the samples before and after cooking, expressed as a percentage. The shear force of the LM samples (in kg/cm^2^) was measured using a TA-XT Plus texturometer coupled with a Warner–Bratzler device. The methods for ash content, WHC, DL, CL, and shear force evaluation were described in detail by Klupsaite et al. [13]. 

The colour coordinates (L*, a*, and b*) were measured using a CIELAB system (Chromameter CR-400, Konica Minolta, Tokyo, Japan). Prior to the colour measurements, the chromameter was calibrated using its white and black calibration tiles. The sample measurements and calibration were performed at the same temperature. The total colour change (ΔE) of the treated samples was compared to that of the control samples after the same treatment time. Then, ΔE was calculated for each sample, according to the method described by Tkacz et al. [70].

### 2.7. Analysis of the Biogenic Amine Contents in the Lamb Meat Samples

The lamb meat sample preparation for the biogenic amine (BA) analysis was caried out using the method described by Ben-Gigirey et al. [71]. The tryptamine, phenylalanine, putrescine, cadaverine, histamine, tyramine, spermidine, and spermine contents were evaluated. The detection limit for BA was 0.1 mg·kg^−1^. The method is described in detail in Appendix A. The method for BA identification was validated in terms of the analytical parameters of accuracy and precision, following the conventional protocols from international guidelines [72,73]. The standard addition (50–200 mg·kg^−1^) recoveries of the samples were as follows: 60–90% for tryptamine, 90–100% for phenylethylamine, 100–105% for putrescine, 95–105% for cadaverine and histamine, 70–100% for tyramine, 95–100% for spermidine, and 90–110% for spermine. The repeatability of the method was tested by analysing the same sample 6–19 times [72,73].

### 2.8. Analysis of the Fatty Acid Profiles of the Lamb Meat Samples

A mixture of chloroform/methanol (2:1 *v*/*v*) was used for the extraction of lipids. The fatty acid methyl ester preparation was carried out according to the method described by Perez-Palacios et al. [74]. The HPLC equipment and chromatographic conditions used for the analysis were described by Klupsaite et al. [19] and are listed in Appendix A.

### 2.9. Analysis of the Overall Acceptability of the Treated Lamb Meat

The overall acceptability of the treated LM samples was evaluated using to the ISO 6658:2017 method [75]. The evaluation was conducted by 30 judges, according to the ISO 8586-1 method [76], using a 10-point scale ranging from 0 (extremely dislike) to 10 (extremely like).

### 2.10. Statistical Analysis

All assays were carried out in triplicate, and the results were expressed as the mean ± standard error (SE). The results were analysed using the SPSS statistical package for Windows V27.0 (SPSS Inc., Chicago, IL, USA, 2020). In order to evaluate the influence of the meat treatments (i.e., the different LAB and pomace used) and its duration, a multivariate analysis of variance (ANOVA) and the Tukey HSD test (as a post hoc test) were performed. The results were considered to be statistically significant at *p* ≤ 0.05. To examine the correlations between the meat quality parameters, the Pearson correlation coefficient was applied.

## 3. Results and Discussion

### 3.1. Microbial and Acidity Parameters of the Prepared Marinades

The acidity and microbial parameters of the prepared marinades are presented in Table 3. In comparison, after 24 h of fermentation, the highest viable LAB count was found in M-AW_Lc_ (6.92 ± 0.04 log_10_ CFU·mL^−1^), while the lowest counts were found in the M-AW_Lu_BC and M-AW_Lu_AP samples (6.15 ± 0.05 log_10_ CFU·mL^−1^ and 6.23 ± 0.04 log_10_ CFU·mL^−1^, respectively). On average, the viable LAB counts of marinades prepared with *L. casei* were 7.5% higher than those of marinades prepared with *L. uvarum*. 

Similar tendencies were also established after 48 h of fermentation. The highest viable LAB counts were found in the M-AW_Lc_ and M-AW_Lu_ marinades (8.82 ± 0.13 log_10_ CFU·mL^−1^ and 8.68 ± 0.05 log_10_ CFU·mL^−1^, respectively), while the lowest was found in M-AW_Lu_BC (7.95 ± 0.09 log_10_ CFU mL^−1^). On average, the viable LAB counts of marinades prepared with *L. casei* were 29.5% higher than those of marinades prepared with *L. uvarum*. The differences in the LAB counts of the marinade samples could be related to the LAB species used. Our previous studies have shown that *L. uvarum* and *L. casei* strains have different growth tendencies in dairy industry by-products [49]; however, both strains reach their highest viable LAB counts after 48 h in selected media [49]. The analysed factors (i.e., fermentation duration, LAB strain, and the type of pomace), as well as their interactions, had significant impacts on the LAB counts of the prepared marinades (*p* ≤ 0.001). The viable LAB counts were lower in marinades with the different pomace products compared to marinades without pomace after the same fermentation time. This could be explained by the fact that, in samples with blackcurrant and apple pomace, pH values lower than 3.45 were found after 48 h of fermentation. Higher acidity could lead to lower LAB counts in samples with AP and BC pomace [77]. Similar tendencies were also found by Du et al. [77], i.e., higher dosages of berry by-products could suspend the growth of LAB, which could be attributed to the high acidity of dairy industry by-products. In our study, the LAB counts of all samples were higher after 48 h of fermentation than 24 h.

After 24 h and 48 h of fermentation, the enterobacteria, yeast, and mould contents in the marinades could not be established (Table 3). 

After 24 h of fermentation, the lowest pH and highest TA values of M-AW_Lc_ (4.13 ± 0.06 and 7.80 ± 0.10 N°, respectively) were found (Table 3). On average, the pH and TA values of marinades prepared with *L. casei* were 3.7% lower and 8.6% higher than those of marinades prepared with *L. uvarum*, respectively.

The different tendencies of the pH and TA values of the marinades after 48 h of fermentation were observed. The lowest pH values were found in the M-AW_Lc_AP, M-AW_Lu_AP, M-AW_Lu_BC, and M-AW_Lc_BC marinades (3.41 ± 0.02, 3.42 ± 0.04, 3.42 ± 0.02, and 3.45 ± 0.03, respectively). In contrast, the highest TA values were found in the M-AW_Lu_AP and M-AW_Lu_BC samples (10.5 ± 0.20 and 10.6 ± 0.2 N°, respectively). The duration of fermentation, LAB strain, and the type of pomace were significant factors in the pH values of the marinades. Additionally, the duration of fermentation, the type of pomace used and their interactions had significant impacts on the TA values of the marinades (*p* < 0.0001). Moreover, a strong negative correlation (*p* < 0.001; R = −0.950) was found between the pH and TA values.

In this experiment, *L. casei* and *L. uvarum* strains were used, which have previously shown a miscellaneous carbohydrate metabolism and a high tolerance to low pH values (viable LAB counts at pH 2.5 after 2 h of incubation were 8.36 ± 0.2 CFU mL^−1^ and 7.55 ± 0.2 CFUmL^−1^, respectively) [33]. We found a strong negative correlation (*p* < 0.001; R = −0.911) between LAB count and pH value, which indicated that by increasing the viable LAB count in a substrate, the pH value decreased. These findings could be explained by the ability of LAB to metabolise carbohydrates into organic acids (mainly lactic acid) via lactic acid fermentation [31]. In dairy products, lactic acid is generated from lactose [78], the level of which in AW varies from 4.2% to 4.9% [79]. Moreover, lactose, glucose, fructose, etc. are carbohydrates that are used for energy production and the subsequent growth of *lactobacilli* strains. According to Bartkiene et al. [32], to improve the biomass growth of LAB strains, whey can be enriched with glucose, sucrose or yeast extracts. According to Zokaityte et al. [49], fermented milk permeated with apple by-products has a high LAB count, which leads to the higher TA and lower pH values of the fermentable substrate. It is also known that berry/fruit by-products include a variety of carbohydrates, minerals, and fibres, which can be used as nutrients for LAB biomass growth and can lead to a higher LAB count in the fermentable substrate [80]. Additionally, carbohydrates are the main carbon sources for LAB biomass growth (especially glucose), although some LAB prefer fructose or lactose [81,82,83].

### 3.2. Antimicrobial Activity of the Developed Marinades 

The antimicrobial activity of the developed marinades against the tested pathogenic and opportunistic bacteria is presented in Table 4. It was established that three out of the six marinades (M-AW_Lc_AP, M-AW_Lu_AP, and M-AW_Lu_BC) inhibited the growth of all of the tested pathogenic and opportunistic strains. The M-AW_Lc_ marinade inhibited six out of the eight tested pathogens (not *Pseudomonas aeruginosa* or *Citrobacter freundii*), and the M-AW_Lc_BC marinade inhibited five out of the eight tested pathogens (not *Pseudomonas aeruginosa, Enterococcus faecalis*, or *Enterococcus faecium*). The M-AW_Lu_ marinade had the lowest antimicrobial activity, which only inhibited the growth of *Proteus mirabilis, Bacillus cereus*, and *Streptococcus mutans*. The highest diameters of inhibition zones (DIZs) against *Salmonella enterica* were observed in the M-AW_Lc_AP and M-AW_Lu_BC marinades (10.3 ± 0.3 mm and 10.30 ± 0.3 mm, respectively). In contrast, the M-AW_Lu_ marinade did not show any antimicrobial activity against this pathogen.

The highest *P. mirabilis* growth suppression was shown by the M-AW_Lc_BC and M-AW_Lu_BC marinades (DIZs of 13.2 ± 0.5 and 13.5 ± 0.3 mm, respectively). On average, antimicrobial activity against *P. aeruginosa* was 10.4 mm and no significant differences were observed between the DIZs of the M-AW_Lc_AP, M-AW_Lu_AP and M-AW_Lu_BC samples. However, the M-AW_Lc_, M-AW_Lc_BC, and M-AW_Lu_ marinades did not show any antimicrobial activity against *P. aeruginosa* or *E. faecium*. The highest *E. faecium* suppression was shown by M-AW_Lu_BC (11.7 ± 0.4 mm). The highest DIZs against *E. faecalis* were shown by the M-AW_Lu_ and M-AW_Lu_AP marinades (12.4 ± 0.2 mm and 12.5 ± 0.4 mm, respectively), while M-AW_Lc_BC did not inhibit the growth of this pathogenic strain. On average, the DIZs against *B. cereus* were 14.2 mm and the highest DIZ was shown by the M-AW_Lc_AP (15.2 ± 0.3 mm) marinade. The highest *S. mutans* growth suppression was demonstrated by the M-AW_Lu_AP marinade, with a DIZ of 16.7 ± 0.4 mm. M-AW_Lu_ and M-AW_Lc_ did not inhibit the growth of *C. freundii*, while the M-AW_Lu_AP and M-AW_Lu_BC marinades showed the highest DIZs against this pathogenic strain (14.3 ± 0.2 mm and 14.2 ± 0.3 mm, respectively). The LAB strain and the type of pomace used, as well as their interactions, had significant effects (*p* ≤ 0.05) on the antimicrobial activity of the samples.

The antimicrobial activity of different ingredients, such as LAB and fruit/berry pomace, are widely debated. It is well known that LAB can inhibit the growth of Gram-positive [32,36,84] and Gram-negative pathogenic bacterial strains [33,63,85,86]. The mechanisms of action LAB are explained by the production of various organic acids (especially lactic acid [87]), which strongly suppress the growth of pathogenic and opportunistic bacteria strains [88]. Due to the lower pH and higher acidity values of the substrate, lactic acid passes through cell membranes and enters cytoplasm, where it then dissociates and lowers the internal pH [87]. This effect effectively kills both Gram-positive and Gram-negative bacteria [87,88]. AP and BC by-products also exert significant effects by suppressing the growth of pathogenic and opportunistic bacteria. These by-products have high quantities of polyphenols (especially anthocyanins), phenolic acid derivatives, flavonols, and proanthocyanins, which all possess antimicrobial activity [36,89,90]. According to Bartkiene et al. [32], biocoats prepared from LAB that are multiplied and lyophilised in dairy industry by-products or lyophilised apple and blackcurrant by-products are very promising antimicrobial ingredients, showing inhibition properties against *S. enterica, S. Typhimurium, E. coli*, *S. aureus*, *B. cereus*, *S. mutans*, *A. baumani*, and *P. multocida* [32]. Additionally, it has been reported that LAB multiplied in milk permeate inhibit *S. eneterica*, *S. Typhimurium*, *E. coli*, *S. aureus*, *B. cereus, Klebsiella pneumoniae,* and *P. marabili* growth better than berry/fruit/vegetable by-products [36,48,49]. The antimicrobial activity of different ingredient combinations is significantly affected by the interactions between different compounds (e.g., LAB strains [32,91,92], the type of by-product [32,37,49,93], and ingredients of plant [85,94,95] or animal origin [96,97,98,99]), as well as the method of valorisation [96,97,100,101]. However, information about the antimicrobial activity of AW and AW in combination with LAB or berry/fruit by-products is scarce. This study showed that the antimicrobial activity of the marinades was related to their composition. Marinades prepared with *L. uvarum* combined with AP and BC pomace demonstrated higher antimicrobial activity than samples prepared with *L. casei*. However, marinades prepared with the *L. uvarum* strain alone displayed lower antimicrobial activity against the tested pathogenic and opportunistic strains.

### 3.3. Influence of the Different Treatments on the Chemical Composition and Technological Parameters of the Lamb Meat Samples

#### 3.3.1. Physicochemical Properties of the Lamb Meat

The physicochemical properties of the LM treated with the different marinades are presented in Table 5. Significant differences were found in the pH values of the marinaded LM between 24 h and 48 h of fermentation (which varied from 5.24 ± 0.06 (LM-AW_Lu_AP) to 6.10 ± 0.06 (LM-AW_Lu_)). For example, after 24 h and 48 h of treatment, reductions in the pH values of LM samples treated with AW fermented with *L. casei* and *L. uvarum* were observed (on average, by 2.1% and 4.4%, respectively); however, the pH values of the other LM samples increased by 3.7%, on average. After 48 h of treatment, the highest pH values were observed in the samples treated with AW fermented with *L. casei* in combination with AP and BC pomace (6.09 ± 0.10 (LM-AW_Lc_AP) and 6.01 ± 0.09 (LM-AW_Lc_BC)). The LAB strain, treatment duration, and the type of pomace used had significant effects (*p* < 0.0001) on the LM pH values; however, the interactions between the analysed factors were not significant. It has been reported that the pH value of meat depends on the ingredients used for marinating, the treatment duration, and the type of meat [20,102]. However, the increase in pH values could also be explained by the microbial metabolism of secondary substrates after the consumption of glucose as, when there is a glucose deficiency, microorganisms have to use secondary nutrients (e.g., amino acids) [19,103,104].

The fat contents of all LM samples in all cases were ambiguous. The same tendencies were also observed when assessing the ash and protein contents of the LM samples treated with the different marinades for 24 and 48 h. The lowest ash content was found in the LM samples treated with AW fermented with L. uvarum alone and in combination with AP and BC pomace for 24 h (0.87% ± 0.04% (LM-AW_Lu_), 0.83% ± 0.02% (LM-AW_Lu_AP), and 0.81% ± 0.04% (LM-AW_Lu_BC)). On average, the ash content of the control sample (C-LM) was 1.28% higher after 24 h and 48 h of treatment. On average, after 48 h of treatment, the ash contents of all treated samples were 17.1% lower compared to that of the control sample (C-LM). On average, after 24 h of treatment, the protein contents of the LM-AW_Lc_AP, LM-AW_Lc_BC, LM-AW_Lu_, and LM-AW_Lu_AP samples were 20.6% higher than those of the other samples. On average, after 48 h, all treated samples had 15.8% higher protein contents than the control sample (C-LM). The highest protein contents after 48 h of treatment were observed in the samples prepared with AW fermented with L. casei alone or in combination with AP and BC pomace (18.9% ± 0.94% (LM-AW_Lc_), 21.0% ± 1.05% (LM-AW_Lc_AP), and 20.3% ± 1.02% (LM-AW_Lc_BC)). Moreover, LAB strain, treatment duration and the type of pomace used, as well as the interactions between these factors, had significant impacts (*p* ≤ 0.05) on the LM fat, ash, and protein contents. The higher protein contents in the marinated LM samples could be explained by the increased protein contents raising the internal osmotic pressure inside the meat, thereby causing the liquid marinade to move from the outside to the inside of the meat rather than the other way around [105]. In addition, fat content is related to cooking loss during the thermal treatment of meat [38]. According to Mozuriene et al. [38], lower fat contents are correlated with higher cooking losses in pork meat samples marinated in fermented potato juice. However, information about how marinades prepared from fermented dairy industry by-products, either alone or in combination with fruit/berry pomace, influence ash and fat contents in meat is scarce. Lastly, the applied marinades had ambiguous effects on the physicochemical properties of the LM samples.

#### 3.3.2. Technological Parameters of the Lamb Meat

The technological parameters of the LM samples treated with the different marinades are presented in Table 6. It was established that after 48 h of treatment, on average, MC increased by 9.7% compared to the C-LM sample. However, no differences were found in MC value between the different groups of treated samples. On average, after 24 h of treatment, most of the samples (not LM-AW_Lc_AP (3.15% ± 0.06%)) showed 41.1% higher WHC values in comparison to the C-LM sample (2.97% ± 0.15%) (Table 6). The highest WHC was observed in the LM-AW_Lu_ sample after 24 h of treatment (5.81% ± 0.11%). On average, WHC decreased by 15.8% after 48 h of treatment compared to 24 h of treatment. However, on average, all treated samples showed 64.1% higher WHC values after 48 h compared to the control sample. Similar tendencies were observed in the CL values (Table 6). On average, CL increased by 63.2% after 24 h in comparison to the control sample. The highest CL values were found in the LM-AW_Lc_, LM-AW_Lc_BC, LM-AW_Lu_, and LM-AW_Lu_AP samples (25.3% ± 1.27%, 23.1% ± 1.16%, 25.5% ± 1.28%, and 22.0% ± 1.09%, respectively). On average, CL increased by 35.4% after 48 h compared to the control sample. After 48 of treatment, the highest CL value was shown by the LM-AW_Lc_AP sample (33.8% ± 1.09%). The interactions between the analysed factors (i.e., treatment duration, LAB strain, and the type of pomace used) had significant impacts on the WHC and CL values (*p* ≤ 0.05) of the LM samples, as well as on MC (*p* < 0.001). 

MC, WHC, and CL are among the most important parameters determining product quality and the safety of fresh meat [105]. It is essential to evaluate the WHC of meat because this parameter is liable for weight deprivation during thermal processing, as well as for raw meat [106]. Reductions in WHC can be induced by the denaturation of the protein components during marination [107]. In this study, a significant (*p* < 0.01) weak positive correlation was found between the WHC and protein contents of the LM samples (R = 0.3935). Additionally, the protein contents and WHC values of all tested samples were higher than those of the control (untreated) sample. According to Mozuriene et al. [38], an increase in water binding can be linked to pH values higher than 5.41 in marinated meat. Our findings were in agreement with those of Sokołowicz et al. [108] and Mozuriene et al. [38], i.e., WHC increased when the pH value of the marinated meat was higher than 5.4. Moreover, a significant (*p* < 0.015) weak positive correlation was found between the WHC and pH values of the LM samples (R = 0.3745). The different components of marinades, the type of meat, and animal breeds have slightly different effects on the technological parameters and chemical composition of meat [19,38,109]. According to Klupsaite et al. [19], on average, *Thymus vulgaris* essential oil can decrease the WHC and CL values of treated LM by 21–63% and 17–35%, respectively, in comparison to the *L. plantarum* strain. According to Sokołowicz et al. [108], marinades based on acetic components (e.g., sour milk and fermented buttermilk) can increase water binding in chicken breast meat due to the swelling of myofibrillar proteins and stronger ionic strengths caused by lower pH values and NaCl contents [110]. However, according to Nour [53], marinades made from sour cherry or plum juice decrease the moisture content and WHC of raw pork meat due to its lower pH value, which is close to the isoelectric point of the meat protein (i.e., pH 5.0–5.2). These results were all induced by the decrease in net charge, the electrostatic repulsion between proteins, and the number of water molecules combining them [53]. Lastly, in most cases, marinades prepared from fermented AW in combination with different plant pomace products improved the MC and WHC of meat, while CL was higher in comparison to the untreated sample. In our study, the marinades prepared from valorised food industry by-products had some positive effects on the LM quality parameters. However, it is important to develop meat products with softer textures due to demand from older consumers [111]. Acidic marinades prepared from food industry by-products together with LAB and their metabolites could lead to improvements in the technological properties of LM. However, further research is needed to establish the appropriate quantities of ingredients and marination time to ameliorate tenderness, chewiness, and sensory parameters, as well as managing any risks associated with the biosafety parameters of LM.

#### 3.3.3. Colour Coordinates, Tenderness, and Overall Acceptability of the Lamb Meat

In addition to the technological parameters (i.e., CL, MC, and WHC) and pH, other critical parameters for consumers include texture, colour coordinates, and sensory factors. Colour is considered to be the most important quality of fresh meat [19,112]. In addition, these parameters play significant roles in improving overall meat quality during the marination process, which must be monitored over the treatment time [38,113]. The colour coordinates, tenderness, and overall acceptability of the LM samples are shown in Table 7. On average, in most cases, the L* values (lightness) of the samples decreased by 20.4% in comparison to C-LM after 24 h of treatment (not the LM-AW_Lu_AP sample). The LM samples treated with *L. casei* alone (LM-AW_Lc_) and in combination with BC pomace (LM-AW_Lc_BC), as well as *L. uvarum* in combination with BC pomace (LM-AW_Lu_BC), possessed the lowest L* coordinates (40.5 ± 0.53 NBS, 39.6 ± 0.58 NBS, and 38.3 ± 0.91 NBS, respectively) after 24 h of treatment. Slightly different tendencies were observed in the L* coordinates after 48 h of LM treatment: on average, the samples with both LAB strains in combination with AP pomace (LM-AW_Lc_AP and LM-AW_Lu_AP) and the control sample (C-LM) were 49.3 NBS lighter. After 48 h of treatment, the LM-AW_Lu_BC sample was the darkest (40.2 ± 0.71 NBS); on average, all samples were 4.5% darker in comparison to after 24 h of treatment. Comparing the a* (redness) results after 24 and 48 h of treatment, the highest a* coordinates were found for the LM-C sample (19.0 ± 0.65 NBS and 15.7 ± 0.35 NBS, respectively) (Table 7). On average, the a* coordinates of all treated samples decreased by 67.3% and 38.7% after 24 h and 48 h of treatment, respectively. After 24 h of treatment, the LM-AW_Lc_BC sample had the lowest a* coordinate (9.22 ± 0.46 NBS); however, after 48 h of treatment, the LM-AW_Lc_, LM-AW_Lc_AP, and LM-AW_Lu_ samples had the lowest a* coordinates (on average, by 10.3 NBS). After 24 h of treatment, the C-LM and LM-AW_Lc_AP samples had the highest b* (yellowness) coordinates (on average, by 12.6 NBS), which were 22% higher than those of the LM-AW_Lc_, LM-AW_Lc_BC, LM-AW_Lu_AP, and LM-AW_Lu_BC samples, on average (Table 7). After 48 h, on average, the b* coordinates of the treated samples were 34.0% lower compared to the control sample (except for the LM-AW_Lu_AP sample). According to Lee et al. [114], the redness (a* coordinate) of pork loin meat can significantly decrease after being marinated with grape pomace; however, this also depends on the other ingredients used in the marinade. The ΔE coefficient determines the total colour difference after evaluating the total changes in the L*, a*, and b* values. On the basis of ΔE, it is possible to predict whether observers or consumers will notice changes in meat colour [70,115]. We found that, in all cases, after 24 and 48 h of marination, the ΔE coefficient was higher than 3, which showed that, in all LM samples, the colour changes would be noticed by consumers. The highest ΔE was observed in the LM-AW_Lc_BC and LM-AW_Lu_BC samples after 24 h of treatment (20.31 and 19.92, respectively). Our study also showed that LAB strain and the type of pomace used for marinade preparation had significant (*p* < 0.0001) influences on the L*, a*, and b* parameters of the LM; however, the effects of treatment duration and the interactions between these factors were not significant. It has been reported that grape pomace powder reduces the redness of pork meat by 40% after 48 h of treatment [114]. Moreover, Klupsaite et al. [19] reported that treatment with *L. plantarum* LUHS135 in combination with *Thymus vulgaris* essential oil reduces a* coordinates in treated lamb meat samples. One of the reasons for this could be that, during treatment at low temperatures (between 6 °C and 4 °C), the redness of lamb meat typically decreases without any additional treatments [116]. We found that there was a significant (*p* < 0.01) weak negative correlation (R = −0.3738) between the pH values and a* coordinates of the LM samples. This is because pH significantly influences the pigments that are accountable for meat colour, which can affect consumer purchasing decisions [117,118,119]. 

The multivariate analysis of variance showed that the LM tenderness was significantly (*p* ≤ 0.05) influenced by treatment duration, LAB strain, and the type of pomace used for the marinade preparation. After 24 h of treatment, the tenderness of the treated LM samples was similar to that of the control sample; however, different tendencies were observed after 48 h. All marinades significantly reduced the shear force values of the LM samples after 48 h of treatment (on average, by 16.2%). According to Hah [120], when the pH of meat decreases during processing (i.e., during marinating [108] or ripening [114]) and it becomes closer to the isoelectric point of meat (pH 5.2), the moisture content of the meat decreases and the shear force increases [114,120]. In this study, we established that the shear force of the treated LM was lower due to the marinated samples having pH values higher than 5.2. According to Mozuriene et al. [38], meat tenderness can be adjusted by altering the pH of the meat via applying marinades prepared with LAB strains (e.g., *L. sakei*, *P acidilactici*, and *P. pentosaceus*). Moreover, there was a significant (*p* < 0.001) strong negative correlation (R = −0.7581) between the tenderness and overall acceptability of the LM samples. On average, the overall acceptability of all marinated LM samples increased by 46.9% compared to that of the control after 24 h of treatment (Table 7). The LM-AW_Lc_BC and LM-AW_Lu_AP samples had the highest overall acceptability after 24 h of treatment (7.23 ± 0.37 and 7.53 ± 0.22, respectively). A similar trend was found after 48 h of treatment; on average, the overall acceptability of all marinated samples increased by 73.2% compared to that of the control. Overall, on average, the overall acceptability of all samples was 35.9% higher after 48 h compared to 24 h. The highest overall acceptability was observed in the LM samples prepared with both strains in combination with AP pomace (LM-AW_Lc_AP and LM-AW_Lu_AP with 9.04 ± 0.43 and 9.43 ± 0.61, respectively). It was also established that the treatment duration, LAB strain, and type of pomace used for marinade preparation, as well as the interactions between these factors, had significant (*p* < 0.001) impacts on the overall acceptability of the LM samples. Our findings are in agreement with those of other researchers, i.e., LAB strains [38,121] and plant-based ingredients (such as grape by-products [114], vegetable extracts [122], and different combinations of ingredients, such as herbs with different oils (pumpkin, olive, vegetable, etc.) [20] and essential oils with lactobacilli [19]) have positive influences on the sensory parameters of marinated meat, and overall acceptability can increase from 15% [20,123] to 31% [124]. Lastly, this study showed that marinades prepared from food industry by-products in combination with selected LAB could improve the texture and overall acceptability of the LM samples. 

### 3.4. The Influence of Different Treatments on the Microbiological and Chemical Parameters of the Lamb Meat Samples

#### 3.4.1. Biogenic Amine Content and Microbiological Parameters of the Lamb Meat

The microbiological characteristics of the LM after 24 and 48 h of treatment are shown in Figure 1a,b, respectively. On average, the LAB counts of all LM samples increased 2.5-fold compared to that of the control after 24 h of treatment (Figure 1a). On average, the LAB counts were 5.1% higher in samples treated with the *L. casei* strain for 24 h compared to those treated with *L. uvarum*. However, the LM samples treated with AW fermented with *L. uvarum* in combination with AP pomace had the highest LAB count after 24 h of treatment (7.53 ± 0.08 log_10_ CFU·g^−1^). Only the C-LM, LM-AW_Lu_AP, and LM-AW_Lu_BC samples contained enterobacteria after 24 h of treatment (2.18 ± 0.05, 1.80 ± 0.19, and 2.30 ± 0.06 log_10_ CFU·g^−1^, respectively). Meanwhile, no Y/M was observed in any samples (Figure 1a). Similar tendencies after 48 h of LM treatment were observed (Figure 1b). The LM-W_Lc_ sample had the highest LAB count (7.72 ± 0.07 log_10_ CFU·g^−1^). When comparing the results from the different treatment durations, the LAB counts of all treated LM samples were 2.8% higher after 48 h, on average; however, the LAB counts of the LM-AW_Lc_AP, LM-AW_Lc_BC, LM-AW_Lu_, LM-AW_Lu_AP, and LM-AW_Lu_BC samples were no different after 48 h of treatment. After 48 h of treatment, the enterobacteria counts in the C-LM, LM-AW_Lu_AP, and LM-AW_Lu_BC samples were 3.66 ± 0.01, 3.30 ± 0.06, and 3.21 ± 0.14 log_10_ CFU·g^−1^, respectively. However, on average, the TEB counts of the treated samples were 11.1% lower than that of the control. Yeast/mould was only found in the C-LM sample after 48 h of treatment (Figure 1b). The treatment duration, LAB strain, and type of pomace used, as well as the interactions between these factors, had significant (*p* < 0.001) effects on the TBC, LAB, and TEB counts of the LM samples. 

This study showed that the different formulations of marinades led to safer LM products. These findings were elucidated by the fact that the ingredients used for the marinades, as well as their combinations, could suppress the growth of undesirable microorganisms. For instance, LAB can excrete or produce bioactive compounds, such as bacteriocins, organic acids, and hydrogen peroxide, which inhibit the growth of other bacteria, yeast, and moulds [125,126,127]. Furthermore, plant ingredients, such as blackcurrants [32,36,46], apples [32,49,128], raspberries [36,37], rowanberries [32,129], and cranberries [130,131], also exert inhibitory effects on undesirable bacteria growth. Our findings agreed with those of Klupsaite et al. [19], who found that treating LM with *lactobacilli* in combination with other plant-based ingredients (i.e., *Thymus vulgaris* essential oil) can improve the safety parameters of the meat. However, data about the influence of combinations of fermented dairy industry by-products and fruit/berry pomace on the microbiological parameters of raw meat are scarce.

The biogenic amine contents of the treated LM samples are presented in Table 8. Tryptamine, putrescine, cadaverine, and histamine were not detected in any of the treated samples after 24 or 48 h of treatment. Phenylethylamine was only found in the LM-AW_Lu_AP and LM-AW_Lu_BC samples after 48 h of treatment (5.80 ± 0.09 and 4.84 ± 0.08 mg·kg^−1^, respectively). Tyramine was only formed in the LM-AW_Lc_BC sample after 24 h (5.58 ± 0.19 mg·kg^−1^). However, after 48 h, tyramine was found in three out of the seven tested samples (LM-AW_Lc_AP, LM-AW_Lc_BC, and LM-AW_Lu_, with tyramine contents of 5.35 ± 0.05, 15.6 ± 0.07, and 20.8 ± 0.20 mg·kg^−1^, respectively). Spermidine was only found in the LM-AW_Lc_BC sample after 24 h of treatment (32.3 ± 0.21 mg·kg^−1^). After 48 h, spermidine contents were also found in the LM-AW_Lc_AP, LM-AW_Lc_BC, LM-AW_Lu_, and LM-AW_Lu_AP samples (23.4 ± 0.24, 22.7 ± 0.17, 37.2 ± 0.25, and 9.48 ± 0.14 mg·kg^−1^, respectively). Spermine was formed in all tested samples. This biogenic amine content of the LM samples ranged from 26.4 ± 0.42 mg·kg^−1^ to 60.6 ± 0.28 mg·kg^−1^ after 24 h of treatment and from 42.7 ± 0.21 to 64.8 ± 0.60 mg·kg^−1^ after 48 h. It is known that spermidine and spermine can be found naturally in fresh meat in quantities of 20–60 mg·kg^−1^ and about 10 mg·kg^−1^, respectively [132,133]. This study showed that only one LM treatment (AW fermented with *L. casei* in combination with blackcurrant pomace) induced a spermine content higher than 60 mg·kg^−1^ (LM-AW_Lc_BC); however, on average, the LM-W_Lc_AP, LM-AW_Lc_BC, and LM-AW_Lu,_ samples contained 2.8-fold more spermidine than 10 mg·kg^−1^. These higher biogenic amine contents could be explained by the fact that the LAB used for the meat treatment had proteolytic activity, as well as the decarboxylase activity of other microbiota, which were not inhibited by the antimicrobial LAB and AP/BC pomace compounds [19,132,133]. Moreover, biogenic amines can be synthesised during meat fermentation using starter cultures, as well as during the technological processing/storage of meat [134,135]. According to EFSA [136], the addition of 10–40 mg of histamine, 5–10 mg of tyramine, 25 mg of tryptamine, or 5 mg of phenylethylamine to foods can cause psychoactive or vasoactive effects on sensitive humans. In this study, the abovementioned biogenic amines were either not established or their concentrations were low in the treated LM samples after 24 h (LM-AW_Lc_BC: 5.58 ± 0.19 mg·kg^−1^). Moreover, histamine is a meat freshness indicator, while high levels of cadaverine and putrescine indicate meat spoilage [137]. In this study, the latter biogenic amines were not found. 

#### 3.4.2. Fatty Acid (FA) Composition of the Lamb Meat

The FA compositions (as a percentage of the total fatty acid content) of the LM samples are presented in Table 9, and the full-scale FA profiles are presented in Appendix A. After 24 h of treatment, the highest total SFA content was found in the LM-AW_Lc_BC sample (59.00% ± 1.21% of the total fatty acid content). The lowest total SFA content was observed in the C-LM sample after 24 h of treatment (55.71% ± 0.86% of the total fatty acid content). Most of the treated LM samples (not C-LM or LM-AW_Lc_BC) showed similar SFA contents. After 48 h of treatment, similar tendencies were also observed and, on average, the total SFA contents of the treated samples were 5.4% higher compared to that of the control. The total MUFA and ω-9 contents of all tested samples were similar; however, slightly different results were observed for the total PUFA contents. After 24 h, the highest total PUFA content was found in the C-LM sample (6.56% ± 0.24% of the total fatty acid content). When comparing the results after the same treatment duration, all treated LM samples had total PUFA contents that were 11.4% lower than that of the C-LM sample, on average. After 48 h of treatment, the total PUFA contents of all tested samples were 9.5% lower in comparison to the results obtained after 24 h of treatment. The highest total PUFA contents were observed in the LM-AW_Lu_ and LM-AW_Lu_BC samples (6.02% ± 0.23% and 6.45% ± 0.25% of the total fatty acid content, respectively). The highest ω-6 FA contents were found in the LM-AW_Lu_BC sample after 24 h and 48 h of treatment (6.09% ± 0.12% and 6.25% ± 0.19% of the total fatty acid content, respectively). After 48 h of treatment, the ω-6 FA contents of four out of the seven treated samples (C-LM, LM-AW_Lc_, LM-AW_Lc_AP, and LM-AW_Lu_AP) decreased (on average, by 15.9% of the total fatty acid content), while the ω-6 FA contents of the other samples (LM-AW_Lc_BC, LM-AW_Lu_, and LM-AW_Lu_BC) were not significantly different compared to the results after 24 h of treatment. No significant differences in ω-9 contents were found between the LM samples. Furthermore, the multivariate analysis of variance indicated that the treatment duration had a significant (*p* < 0.0001) influence on the total PUFA content. In contrast, the LAB strain had a significant influence (*p* < 0.0001) on the total SFA and PUFA contents, while the interactions between these factors was significant (*p* < 0.0001). Moreover, the interaction between LAB strain and the type of pomace used had a significant impact on the total PUFA contents of the tested samples, as did the interaction among all tested factors.

LM is a source of essential FAs, especially linolenic (C18:3 cis-6,9,15), eicosapentaenoic (C20:5 *n*3), and docosahexaenoic (C22: 6 𝑛 − 3) acids, which have positive influences on cardiovascular disease prevention [138,139]. Moreover, the ratio of PUFA/SFA is very important and can vary from 0.06 [14,19,140,141] to 0.59 [142,143,144] in LM, according to various researchers. Our findings are in agreement with the abovementioned results because we found that the PUFA/SFA ratio in the treated LM samples was 0.1. However, the FA compositions of different meats (lamb, chicken, pork, etc.) depend on the type of farming, feed composition, breed, and animal age [140,141,143,145,146,147,148]. According to Klupsaite et al. [19], FA composition also depends on the ingredients used for processing; for instance, treatment with sunflower oil and *Thymus vulgaris* essential oil alone or in combination with *L. plantarum* has a significant effect (*p* ≤ 0.05) on the FA profile of LM. LAB can also cause changes in the FA composition of meat due to their lipolytic activity, as well as their capacity to preclude the oxidation of unsaturated free FAs [19,149]. In conclusion, the different marinades and treatment durations had miscellaneous effects on the FA composition of the LM.

## 4. Conclusions

This study showed that 48 h of treatment should be recommended for marinades because, in all cases (except for the marinade prepared from acid whey fermented with *L. uvarum* in combination with BC pomace (M-AW_Lu_BC)), the viable LAB cell counts were higher than 8.0 log_10_ CFU·mL^−1^ and the pH values were lower than 3.74. The marinades prepared with *L. uvarum* in combination with AP and BC pomace had slightly higher antimicrobial activity and inhibited the growth of all tested pathogenic and opportunistic strains. Moreover, marinades made from acid whey fermented with *L. casei* alone or in combination with AP/BC pomace or acid whey fermented with *L. uvarum* alone or in combination with AP/BC pomace could be recommended for LM treatment to improve technological parameters (MC, WHC, texture, etc.). The overall acceptability of the LM improved after 48 h of treatment. Furthermore, the marinade made from acid whey fermented with *L. casei* in combination with AP pomace improved the biosafety parameters of the LM. Additionally, histamine, cadaverine, putrescine, tryptamine, and phenylethylamine were not formed in any of the treated samples, and their spermine contents were lower than 60 mg·kg^−1^. Lastly, food industry by-products combined with selected LAB strains could be promising ingredients for marinades that aim to improve the biosafety and quality parameters, as well as the overall acceptability, of LM. The best formulations were LM-AW_Lc_AP and LM-AW_Lu_AP. 

## Figures and Tables

**Figure 1 foods-12-01391-f001:**
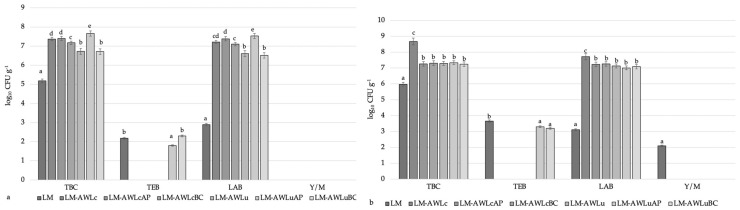
The influence of the different treatments on the microbial parameters of the LM after (**a**) 24 h and (**b**) 48 h. ^a−e^ Mean values with different superscript letters between columns are significantly different (*p* ≤ 0.05); data are expressed as the mean values (*n* = 3) ± standard error (SE); C, control; LM, lamb meat; AW, acid whey; Lc, *Lacticaseibacillus casei*; Lu, *Liquorilactobacillus uvarum;* AP, freeze-dried apple pomace; BC, freeze-dried blackcurrant pomace.

**Table 1 foods-12-01391-t001:** The ingredients used for the formulation of the marinades.

Sample	LAB (%)		AW (mL)	Freeze-Dried AP Pomace (g)	Freeze-Dried BC Pomace (g)
	*L. casei*	*L. uvarum*			
M-AW_Lc_	3	-	120	-	-
M-AW_Lc_AP		-		3.00	
M-AW_Lc_BC		-			3.00
M-AW_Lu_	-	3		-	-
M-AW_Lu_AP	-			3.00	-
M-AW_Lu_BC	-				3.00

M, marinade; AW, acid whey; Lc, *Lacticaseibacillus casei* LUHS210; Lu, *Liquorilactobacillus uvarum* LUHS245; AP, freeze-dried apple pomace; BC, freeze-dried blackcurrant pomace.

**Table 2 foods-12-01391-t002:** The lamb meat treatment formulations.

Samples	Lamb Meat (g)	AW_Lc_ (mL)	AW_Lu_ (mL)	Freeze-Dried AP Pomace (g)	Freeze-Dried BC Pomace (g)
LM-C	400	-	-	-	-
LM-AW_Lc_		120	-	-	-
LM-AW_Lc_AP			-	3.00	-
LM-AW_Lc_BC			-	-	3.00
LM-AW_Lu_		-	120	-	-
LM-AW_Lu_AP		-		3.00	-
LM-AW_Lu_BC		-		-	3.00

LM, lamb meat; C, control; AW, acid whey; Lc, *Lacticaseibacillus casei;* Lu, *Liquorilactobacillus uvarum*; AP, freeze-dried apple pomace; BC, freeze-dried blackcurrant pomace.

**Table 3 foods-12-01391-t003:** The acidity and microbial parameters of the marinades.

Parameter	Marinade					
	M-AW_Lc_	M-AW_Lc_AP	M-AW_Lc_BC	M-AW_Lu_	M-AW_Lu_AP	M-AW_Lu_BC
	Log_10_ CFU mL^−1^					
	After 24 h of fermentation					
TBC	7.34 ± 0.06 ^c^	6.79 ± 0.05 ^ab^	6.64 ± 0.14 ^ab^	6.80 ± 0.08 ^b^	6.67 ± 0.11 ^ab^	6.47 ± 0.12 ^a^
LAB	6.92 ± 0.04 ^e^	6.78 ± 0.06 ^d^	6.51 ± 0.05 ^c^	6.41 ± 0.03 ^b^	6.23 ± 0.04 ^a^	6.15 ± 0.05 ^a^
TEB	nd	nd	nd	nd	nd	nd
M/Y	nd	nd	nd	nd	nd	nd
	After 48 h of fermentation					
TBC	8.94 ± 0.09 ^d^	8.57 ± 0.07 ^b^	8.55 ± 0.04 ^b^	8.74 ± 0.06 ^c^	8.26 ± 0.10 ^a^	8.16 ± 0.06 ^a^
LAB	8.82 ± 0.13 ^d^	8.42 ± 0.15 ^c^	8.47 ± 0.06 ^c^	8.68 ± 0.05 ^d^	8.15 ± 0.07 ^b^	7.95 ± 0.09 ^a^
TEB	nd	nd	nd	nd	nd	nd
M/Y	nd	nd	nd	nd	nd	nd
	Acidity Parameters					
	After 24 h of fermentation					
pH	4.13 ± 0.06 ^a^	4.23 ± 0.03 ^b^	4.25 ± 0.02 ^b^	4.32 ± 0.02 ^c^	4.41 ± 0.03 ^d^	4.36 ± 0.04 ^cd^
TA (N°)	7.80 ± 0.10 ^d^	7.40 ± 0.10b ^c^	7.50 ± 0.10 ^c^	7.20 ± 0.10 ^b^	6.80 ± 0.20 ^a^	6.90 ± 0.10 ^a^
	After 48 h of fermentation					
pH	3.61 ± 0.03 ^b^	3.41 ± 0.02 ^a^	3.45 ± 0.03 ^a^	3.74 ± 0.01 ^c^	3.42 ± 0.04 ^a^	3.42 ± 0.02 ^a^
TA (N°)	8.60 ± 0.10 ^a^	9.70 ± 0.10 ^bc^	9.80 ± 0.20 ^bc^	9.50 ± 0.10 ^b^	10.5 ± 0.20 ^d^	10.6 ± 0.2 ^d^

^a–e^ Mean values with different superscript letters between rows are significantly different (*p* ≤ 0.05); data are expressed as the mean values (*n* = 3) ± standard error (SE); nd, not detected; LAB, lactic acid bacteria count; TBC, total bacteria count; TEB, total enterobacteria count; M/Y, mould and yeast count; CFU, colony-forming units; TA, titratable acid; M, marinade; AW, acid whey; Lc, *Lacticaseibacillus casei;* Lu, *Liquorilactobacillus uvarum***;** AP, freeze-dried apple pomace; BC, freeze-dried blackcurrant pomace.

**Table 4 foods-12-01391-t004:** The antimicrobial activity of the prepared marinades.

Inhibition Zone (mm)								
Pathogenic Bacteria Strain								
	*Salmonella* *enterica*	*Pseudomonas aeruginosa*	*Proteus* *mirabilis*	*Enterococcus faecalis*	*Enterococcus faecium*	*Bacillus* *cereus*	*Streptococcus mutans*	*Citrobacter freundii*
M-AW_Lc_	9.20 ± 0.3 ^a^	nd	10.3 ± 0.1 ^a^	10.1 ± 0.3 ^a^	9.40 ± 0.2 ^a^	14.3 ± 0.2 ^b^	15.2 ± 0.4 ^c^	nd
M-AW_Lc_AP	10.3 ± 0.3 ^b^	10.1 ± 0.4 ^a^	12.1 ± 0.3 ^c^	9.90 ± 0.1 ^a^	9.80 ± 0.4 ^a^	15.2 ± 0.3 ^c^	14.3±0.2 ^b^	10.1 ± 0.2 ^a^
M-AW_Lc_BC	8.90 ± 0.4 ^a^	nd	13.2 ± 0.5 ^d^	nd	nd	14.3 ± 0.1 ^b^	14.3 ± 0.3 ^b^	11.4 ± 0.3 ^b^
M-AW_Lu_	nd	nd	11.4 ± 0.2 ^b^	12.4 ± 0.2 ^c^	nd	13.4 ± 0.1 ^a^	13.2 ± 0.4 ^a^	nd
M-AW_Lu_AP	9.40 ± 0.3 ^a^	10.6 ± 0.2 ^a^	12.2 ± 0.1 ^c^	12.5 ± 0.4 ^c^	10.5 ± 0.3 ^b^	14.3 ± 0.3 ^b^	16.7 ± 0.4 ^d^	14.3 ± 0.2 ^c^
M-AW_Lu_BC	10.30 ± 0.3 ^b^	10.6 ± 0.3 ^a^	13.5 ± 0.3 ^d^	11.8 ± 0.2 ^b^	11.7 ± 0.4 ^c^	13.7 ± 0.2 ^a^	15.5 ± 0.1 ^c^	14.2 ± 0.3 ^c^

^a–d^ Mean values with different superscript letters between columns are significantly different (*p* ≤ 0.05); data are expressed as the mean values (*n* = 3) ± standard error (SE); nd, not detected; M, marinade; AW, acid whey; Lc, *Lacticaseibacillus casei;* Lu, *Liquorilactobacillus uvarum*; AP, freeze-dried apple pomace; BC, freeze-dried blackcurrant pomace.

**Table 5 foods-12-01391-t005:** The chemical composition of the lamb meat samples treated with the different marinades.

Sample	Physicochemical Properties							
	pH		Fat Content (%)		Ash Content (%)		Protein Content (%)	
	Duration of Treatment (h)							
	24	48	24	48	24	48	24	48
C-LM	5.40 ± 0.05 ^b^	5.51 ± 0.07 ^a^	10.5 ± 0.05 ^c^	9.62 ± 0.12 ^c^	1.32 ± 0.09 ^c^	1.24 ± 0.11 ^d^	17.1 ± 0.85 ^a^	16.3 ± 0.32 ^a^
LM-AW_Lc_	5.89 ± 0.09 ^c^	5.77 ± 0.09 ^ab^	11.1 ± 0.25 ^d^	6.79 ± 0.14 ^a^	0.97 ± 0.05 ^b^	1.13 ± 0.06 ^c^	17.7 ± 0.88 ^a^	18.9 ± 0.94 ^c^
LM-AW_Lc_AP	5.72 ± 0.08 ^c^	6.09 ± 0.10 ^c^	10.6 ± 0.25 ^c^	6.90 ± 0.25 ^a^	0.91 ± 0.05 ^ab^	1.10 ± 0.07 ^c^	20.6 ± 1.03 ^b^	21.0 ± 1.05 ^c^
LM-AW_Lc_BC	5.87 ± 0.09 ^d^	6.01 ± 0.09 ^bc^	7.65 ± 0.28^a^	6.99 ± 0.15 ^a^	0.98 ± 0.05 ^b^	1.07 ± 0.05 ^c^	19.4 ± 0.97^b^	20.3 ± 1.02 ^c^
LM-AW_Lu_	6.10 ± 0.06 ^e^	5.83 ± 0.09 ^b^	9.37 ± 0.17 ^b^	7.28 ± 0.06 ^b^	0.87 ± 0.04 ^a^	1.03 ± 0.05 ^c^	21.4 ± 1.07 ^b^	17.7 ± 0.89 ^b^
LM-AW_Lu_AP	5.24 ± 0.06 ^a^	5.57 ± 0.08^a^	11.1 ± 0.24 ^d^	10.5 ± 0.18 ^d^	0.83 ± 0.02 ^a^	0.70 ± 0.02 ^a^	20.8 ± 1.04 ^b^	17.5 ± 0.87 ^b^
LM-AW_Lu_BC	5.62 ± 0.08 ^c^	5.69 ± 0.07 ^a^	13.4 ± 0.18 ^e^	12.7 ± 0.23 ^e^	0.81 ± 0.04 ^a^	0.81 ± 0.04 ^b^	16.7 ± 0.84 ^a^	17.9 ± 0.89 ^b^

^a–e^ Mean values with different superscript letters between columns are significantly different (*p* ≤ 0.05); data are expressed as the mean values (*n* = 3) ± standard error (SE); C, control (without treatment); LM, lamb meat; AW, acid whey; Lc, *Lacticaseibacillus casei;* Lu, *Liquorilactobacillus uvarum*; AP, freeze-dried apple pomace; BC, freeze-dried blackcurrant pomace.

**Table 6 foods-12-01391-t006:** The technological parameters of the lamb meat samples treated with the different marinades.

	Technological Parameters (%)							
	DM		MC		WHC		CL	
	Duration of Treatment (h)							
	24	48	24	48	24	48	24	48
C-LM	33.4 ± 1.13 ^b^	32.9 ± 0.95 ^b^	69.7 ± 1.15 ^a^	65.8 ± 1.22 ^a^	2.97 ± 0.15 ^a^	2.42 ± 0.18 ^a^	13.6 ± 0.56 ^a^	21.9 ± 0.56 ^a^
LM-AW_Lc_	29.8 ± 1.49 ^b^	26.8 ± 1.34 ^a^	72.9 ± 1.10 ^a^	73.2 ± 0.68 ^b^	5.03 ± 0.0 8^c^	4.81 ± 0.23 ^f^	25.3 ± 1.27 ^d^	23.0 ± 1.51 ^a^
LM-AW_Lc_AP	26.6 ± 1.33 ^a^	27.0 ± 1.35 ^a^	74.2 ± 1.05 ^a^	72.5 ± 1.35 ^b^	3.15 ± 0.06 ^a^	2.91 ± 0.15 ^b^	19.7 ± 0.99 ^c^	33.8 ± 1.09 ^c^
LM-AW_Lc_BC	26.0 ± 1.30 ^a^	26.3 ± 1.32 ^a^	72.3 ± 0.98 ^a^	71.8 ± 0.68 ^b^	5.50 ± 0.0 ^d^	4.74 ± 0.06 ^f^	23.1 ± 1.16 ^d^	29.3 ± 1.17 ^b^
LM-AW_Lu_	31.6 ± 1.58 ^b^	26.1 ± 1.30 ^a^	71.4 ± 0.53 ^a^	71.5 ± 1.33 ^b^	5.81 ± 0.11 ^e^	4.43 ± 0.17 ^e^	25.5 ± 1.28 ^d^	29.6 ± 1.48 ^b^
LM-AW_Lu_AP	42.7 ± 2.14 ^c^	35.3 ± 1.76 ^c^	71.6 ± 0.71 ^a^	72.0 ± 1.50 ^b^	4.81 ± 0.15 ^c^	3.59 ± 0.13 ^d^	22.0 ± 1.09 ^d^	32.7 ± 1.24 ^bc^
LM-AW_Lu_BC	31.1 ± 1.56 ^b^	38.5 ± 1.93 ^c^	72.2 ± 0.61 ^a^	71.9 ± 0.95 ^b^	3.92 ± 0.21 ^b^	3.36 ± 0.07 ^c^	17.6 ± 0.88 ^b^	29.6 ± 1.48 ^b^

^a–f^ Mean values with different superscript letters between columns are significantly different (*p* ≤ 0.05); data are expressed as the mean values (*n* = 3) ± standard error (SE); C, control (without treatment); LM, lamb meat; AW, acid whey; Lc, *Lacticaseibacillus casei;* Lu, *Liquorilactobacillus uvarum*; AP, freeze-dried apple pomace; BC, freeze-dried blackcurrant pomace; DM, dry matter; MC, moisture content; WHC, water holding capacity; CL, cooking loss.

**Table 7 foods-12-01391-t007:** The colour coordinates, tenderness, and overall acceptability of the lamb meat samples treated with the different marinades.

	Colour Coordinates								Shear Force (kg·cm^−2^)		Overall Acceptability Score	
	L*		a*		b*		ΔE					
	Duration of Treatment (h)											
	24	48	24	48	24	48	24	48	24	48	24	48
C_LM_	51.9 ±1.29 ^d^	48.9 ±0.96 ^e^	19.0 ± 0.65 ^f^	15.7 ± 0.35 ^d^	12.1 ± 0.60 ^b^	10.5 ± 0.14 ^d^	0	0	1.75 ± 0.09 ^a^	1.83 ± 0.12 ^c^	3.42 ± 0.33 ^a^	2.13 ± 0.20 ^a^
LM-AW_Lc_	40.5 ± 0.53 ^a^	45.3 ± 0.27 ^c^	6.60 ± 0.33 ^c^	9.93 ± 0.49 ^a^	10.1 ± 0.51 ^a^	7.27 ± 0.36 ^b^	16.95	7.53	1.70 ± 0.10 ^a^	1.64 ± 0.05 ^b^	5.68 ± 0.24 ^b^	6.02 ± 0.43 ^b^
LM-AW_Lc_AP	43.3 ± 0.47 ^b^	49.3 ± 0.76 ^e^	7.60 ± 0.18 ^d^	10.2 ± 0.51 ^a^	13.0 ± 0.65 ^b^	9.37 ± 0.47 ^c^	14,30	5.63	1.61 ± 0.09 ^a^	1.54 ± 0.03 ^a^	6.07 ± 0.46 ^b^	9.04 ± 0.43 ^d^
LM-AW_Lc_BC	39.6 ± 0.58 ^a^	41.9 ± 0.49 ^b^	2.94 ± 0.15 ^a^	11.6 ± 0.18 ^b^	10.3 ± 0.52 ^a^	4.86 ± 0.24 ^a^	20.31	9.88	1.59 ± 0.05 ^a^	1.52 ± 0.04 ^a^	7.23 ± 0.37 ^c^	8.12 ± 0.15 ^c^
LM-AW_Lu_	44.6 ± 0.23 ^c^	46.8 ± 0.34 ^d^	6.48 ± 0.32 ^c^	10.8 ± 0.34 ^a^	11.6 ± 0.43 ^ab^	7.63 ± 0.38 ^b^	14.50	6.25	1.69 ± 0.07 ^a^	1.55 ± 0.03 ^a^	5.99 ± 0.61 ^b^	6.97 ± 0.31 ^b^
LM-AW_Lu_AP	50.6 ± 0.53 ^d^	49.8 ± 0.49 ^e^	9.22 ± 0.46 ^e^	11.7 ± 0.39 ^b^	9.74 ± 0.49 ^a^	10.3 ± 0.22 ^d^	10.14	4.10	1.61 ± 0.10 ^a^	1.47 ± 0.06 ^a^	7.53 ± 0.22 ^c^	9.43 ± 0.61 ^d^
LM-AW_Lu_BC	38.3 ± 0.91 ^a^	40.2 ± 0.71 ^a^	4.49 ± 0.22 ^b^	13.1 ± 0.66 ^c^	11.0 ± 0.55 ^a^	5.54 ± 0.28 ^a^	19.92	10.35	1.64 ± 0.11 ^a^	1.48 ± 0.04 ^a^	6.14 ± 0.19 ^b^	8.16 ± 0.28 ^c^

^a–f^ Mean values with different superscript letters between columns are significantly different (*p* ≤ 0.05); data are expressed as the mean values (*n* = 3) ± standard error (SE); C, control (without treatment); LM, lamb meat; AW, acid whey; Lc, *Lacticaseibacillus casei;* Lu, *Liquorilactobacillus uvarum***;** AP, freeze-dried apple pomace; BC, freeze-dried blackcurrant pomace; L*, lightness; a*, redness; b*, yellowness.

**Table 8 foods-12-01391-t008:** The biogenic amine contents of the LM samples treated with the different marinades.

Samples	Biogenic Amines (mg·kg^−1^)							
	Phenylethylamine		Tyramine		Spermidine		Spermine	
	Duration of Treatment (h)							
	24	48	24	48	24	48	24	48
C-LM	nd	nd	nd	nd	nd	nd	42.3 ± 0.55 ^d^	54.2 ± 0.15 ^e^
LM-AW_Lc_	nd	nd	nd	nd	nd	nd	43.7 ± 0.29 ^e^	53.6 ± 0.17 ^d^
LM-W_Lc_AP	nd	nd	nd	5.35 ± 0.05 ^a^	nd	23.4 ± 0.24 ^c^	41.2 ± 0.31 ^c^	51.8 ± 0.19 ^c^
LM-AW_Lc_BC	nd	nd	5.58 ± 0.19	15.6 ± 0.0 ^b^	32.3 ± 0.21	22.7 ± 0.17 ^b^	55.2 ± 0.61 ^f^	64.8 ± 0.60 ^g^
LM-AW_Lu_	nd	nd	nd	20.8 ± 0.20 ^c^	nd	37.2 ± 0.25 ^d^	60.6 ± 0.28 ^g^	57.8 ± 0.58 ^f^
LM-AW_Lu_AP	nd	5.80 ± 0.0 ^b^	nd	nd	nd	9.48 ± 0.14 ^a^	26.4 ± 0.42 ^a^	42.7 ± 0.21 ^a^
LM-AW_Lu_BC	nd	4.84 ± 0.08 ^a^	nd	nd	nd	nd	36.8 ± 0.16 ^b^	50.9 ± 0.20 ^b^

^a–g^ Mean values with different superscript letters between columns are significantly different (*p* ≤ 0.05); data are expressed as the mean values (*n* = 3) ± standard error (SE); C, control; LM, lamb meat; AW, acid whey; Lc, *Lacticaseibacillus casei*; Lu, *Liquorilactobacillus uvarum*; AP, freeze-dried apple pomace; BC, freeze-dried blackcurrant pomace.

**Table 9 foods-12-01391-t009:** The fatty acid compositions (as a percentage of the total fatty acid content) and the major fatty acids of the LM samples.

Sample	Fatty Acid Composition (% of Total Fatty Acid Content)											
	Total SFAs		Total MUFAs		Total PUFAs		Omega-3 (ω-3)		Omega-6 (ω-6)		Omega-9 (ω-9)	
	Duration of Treatment (h)											
	24	48	24	48	24	48	24	48	24	48	24	48
C-LM	55.71 ±0.86 ^a^	54.61 ± 0.95 ^a^	37.63 ± 1.19 ^a^	37.15 ± 1.04 ^a^	6.56 ± 0.14 ^c^	5.51 ± 0.11 ^b^	0.25 ± 0.070 ^a^	0.23 ± 0.009 ^d^	6.31 ± 0.11 ^b^	5.28 ± 0.12 ^b^	34.67 ± 0.43 ^a^	34.23 ± 0.65 ^a^
LM-AW_Lc_	57.87 ± 1.06 ^b^	58.78 ± 0.74 ^b^	36.22 ± 0.64 ^a^	36.36 ± 0.34 ^a^	5.83 ± 0.18 ^b^	4.63 ± 0.15 ^a^	0.20 ± 0.050 ^a^	0.17 ± 0.010 ^b^	5.63 ± 0.12 ^c^	4.46 ± 0.10 ^a^	34.25 ± 0.65 ^a^	34.04 ± 0.71 ^a^
LM-AW_Lc_AP	57.33 ± 0.94 ^b^	57.99 ± 1.07 ^b^	37.09 ± 1.04 ^a^	36.91 ± 0.86 ^a^	5.98 ± 0.12 ^b^	4.82 ± 0.20 ^a^	0.30 ± 0.050 ^a^	0.39 ± 0.010 ^e^	5.68 ± 0.11 ^c^	4.43 ± 0.09 ^a^	34.34 ± 0.54 ^a^	34.6 ± 0.89 ^a^
LM-AW_Lc_BC	59.00 ± 1.21 ^c^	58.19 ± 0.74 ^b^	37.00 ± 0.83 ^a^	36.61 ± 1.00 ^a^	4.42 ± 0.08 ^a^	4.78 ± 0.19 ^a^	0.27 ± 0.011 ^a^	0.22 ± 0.009 ^d^	4.15 ± 0.19 ^a^	4.56 ± 0.21 ^a^	34.96 ± 1.02 ^a^	34.45 ± 0.89 ^a^
LM-AW_Lu_	57.32 ± 0.75 ^b^	56.48 ± 0.87 ^b^	36.50 ± 0.41 ^a^	36.66 ± 0.78 ^a^	6.03 ± 0.11 ^b^	6.02 ± 0.23 ^c^	0.34 ± 0.012 ^a^	0.20 ± 0.008 ^c^	5.69 ± 0.19 ^c^	5.82 ± 0.11 ^c^	34.17 ± 0.70 ^a^	34.75 ± 0.61 ^a^
LM-AW_Lu_AP	56.71 ± 0.87 ^ab^	57.37 ± 0.91 ^b^	37.08 ± 0.97 ^a^	37.03 ± 0.75 ^a^	6.13 ± 0.17 ^b^	5.15 ± 0.17 ^a^	0.31 ± 0.070 ^a^	0.14 ± 0.004 ^a^	5.82 ± 0.11 ^c^	5.01 ± 0.22 ^b^	34.60 ± 0.54 ^a^	34.77 ± 0.85 ^a^
LM-AW_Lu_BC	56.41 ± 0.93 ^b^	56.46 ± 0.74 ^b^	37.13 ± 1.10 ^a^	36.73 ± 0.84 ^a^	6.30 ± 0.10 ^b^	6.45 ± 0.25 ^c^	0.21 ± 0.021 ^a^	0.20 ± 0.011 ^d^	6.09 ± 0.12 ^d^	6.25 ± 0.19 ^d^	34.70 ± 0.66 ^a^	34.58 ± 0.39 ^a^
**Saturated Fatty Acids (SFAs)**					**Monounsaturated Fatty Acids (MUFAs)**				**Polyunsaturated Fatty Acids (PUFAs)**			
	Palmitic C16:0		Stearic C18:0		Oleic C18:1 cis-9		Palmitoleic C16:1 cis-9		Linoleic C18:2 cis-9,12		**γ**-Linolenic C18:3 cis-6,9,12	
	24	48	24	48	24	48	24	48	24	48	24	48
C-LM	22.59 ± 0.75 ^a^	22.00 ± 0.66 ^a^	24.10 ± 0.43 ^a^	23.80 ± 0.32 ^a^	32.45 ± 0.84 ^a^	31.98 ± 0.72 ^a^	2.82 ± 0.17 ^b^	2.81 ± 0.11 ^b^	3.70 ± 0.23 ^b^	3.21 ± 0.11 ^a^	2.49 ± 0.13 ^c^	1.99 ± 0.12 ^b^
LM-AW_Lc_	24.54 ± 0.81 ^a^	24.83 ± 0.53 ^b^	24.07 ± 0.20 ^a^	24.40 ± 0.25 ^a^	32.87 ± 1.25 ^a^	32.82 ± 0.94 ^a^	1.86 ± 0.16 ^a^	2.13 ± 0.09 ^a^	3.34 ± 0.19 ^a^	2.84 ± 0.14 ^a^	2.15 ± 0.150 ^b^	1.44 ± 0.16 ^a^
LM-AW_Lc_AP	24.52 ± 0.85 ^a^	24.43 ± 0.45 ^b^	24.41 ± 0.64 ^ab^	25.75 ± 0.31 ^c^	33.16 ± 1.02 ^a^	33.43 ± 0.36 ^a^	2.62 ± 0.23 ^b^	2.19 ± 0.13 ^a^	3.24 ± 0.16 ^a^	2.88 ± 0.13 ^a^	2.12 ± 0.144 ^b^	1.20 ± 0.20 ^a^
LM-AW_Lc_BC	24.81 ± 0.77 ^a^	24.97 ± 0.64 ^b^	24.73 ± 0.19 ^a^	24.83 ± 0.24 ^b^	33.78 ± 0.54 ^a^	33.29 ± 0.47 ^a^	1.97 ± 0.24 ^a^	2.09 ± 0.10 ^a^	2.83 ± 0.18 ^a^	2.96 ± 0.22 ^a^	1.14 ± 0.23 ^a^	1.43 ± 0.13 ^a^
LM-AW_Lu_	24.67 ± 0.54 ^a^	23.94 ± 0.41 ^b^	24.67 ± 0.29 ^a^	24.29 ± 0.18 ^a^	32.45 ± 0.69 ^a^	32.45 ± 0.99 ^a^	2.24 ± 0.17 ^ab^	1.86 ± 0.14 ^a^	3.14 ± 0.23 ^a^	3.45 ± 0.36 ^b^	2.37 ± 0.12 ^bc^	2.22 ± 0.23 ^b^
LM-AW_Lu_AP	23.28 ± 0.57 ^a^	24.45 ± 0.69 ^b^	24.64 ± 0.24 ^a^	24.71 ± 0.37 ^b^	32.77 ± 0.88 ^a^	33.19 ± 1.12 ^a^	2.33 ± 0.18 ^b^	2.17 ± 0.20 ^a^	3.20 ± 0.30 ^a^	2.99 ± 0.18 ^a^	2.46 ± 0.17 ^c^	1.85 ± 0.15 ^b^
LM-AW_Lu_BC	23.38 ± 0.64 ^a^	23.32 ± 0.41 ^b^	24.04 ± 0.22 ^a^	24.18 ± 0.19 ^a^	32.74 ± 0.82 ^a^	32.59 ± 0.93 ^a^	2.33 ± 0.11 ^b^	2.03 ± 0.09 ^a^	3.64 ± 0.15 ^b^	3.87 ± 0.18 ^b^	2.29 ± 0.19 ^b^	2.26 ± 0.17 ^b^

^a–e^ Mean values with different superscript letters between columns are significantly different (*p* ≤ 0.05); data are expressed as the mean values (*n* = 3) ± standard error (SE); LM, lamb meat; AW, acid whey; Lc, *Lacticaseibacillus casei;* Lu, *Liquorilactobacillus uvarum*; AP, freeze-dried apple pomace; BC, freeze-dried blackcurrant pomace; SFAs, saturated fatty acids; MUFAs, monounsaturated fatty acids; PUFAs, polyunsaturated fatty acids.

## Data Availability

The data are available from the corresponding author upon reasonable request.

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
