# Peer review of "Effects of Marinades Prepared from Food Industry By-Products on Quality and Biosafety Parameters of Lamb Meat"

_foods, 2023, doi:10.3390/foods12071391_

Round 1

Reviewer 1 Report

Title: Effect of Marinade Based Prepared from Valorised Food Industry By-products on Quality and Biosafety Parameters of Lamb Meat

The manuscript “Effect of Marinade Based Prepared from Valorised Food Industry By-products on Quality and Biosafety Parameters of Lamb Meat” aimed to develop marinade formulas based on dairy, berries and fruit industry by-products and applied them to lamb meat treatment to improve meat safety and quality characteristics. It is well written article with some interesting findings; however, there are some corrections to be done:

Its my suggestion to remove the word “valorised” from the title to make it simple for the readers to understand.

Line 23: Its “combined” not “combned.

Line 23: How combinations were made?

Line 24: How authors have selected this criteria? Is it available in the literature? “The most appropriate marinades fermentation time was selected according to pH and viable LAB count.”

Line 29: What is the meanin of “M” in M-AWLuBP. I think authors should abbreviate it before using in order to make it clear of the meaning for the reader.

Line 41: In introduction part, authors should discuss about the wastage and safety aspects of lamb meat in whole world. I would suggest the authors to read the following articles and discuss it in first paragraph of the introduction:

·       Food Loss and Waste in Meat Sector—Why the Consumption Stage Generates the Most Losses? (https://doi.org/10.3390/su13116227)

·       Lamb From Farm to Table (https://www.fsis.usda.gov/food-safety/safe-food-handling-and-preparation/meat/lamb-farm-table)

Line 56: In second paragraph of the introduction, authors should also discuss about usage of synthetic compounds in the lamb meat and its unacceptability issue by the consumers?

Line 133: Give reference?

Line 142: It would be better, if authors add “Figure 1: The principal scheme of the experiment…” in the supplementary file.

Line 147: When you said LM was treated with… then no need to write LM along the treatments such as AWLc….

Line 154: To make it clear for the readers, authors should add borders in the Table 2.

Line 100: At what time postmortem the lamb meat was collected?

Line 157: How about calibration of the pH meter?

Line 190: Calibration of Chromameter CR-400?

Line 304: I would suggest to improve the discussion part by indicating the active ingredients present in fruit/berry pomace.

Line 435: Please calculate the change in color (ΔE) values and incorporate and discuss the results.

Line 410: Authors should suggest some guidelines for future research, i.e., which aspect should be focused for future research.

English grammar and sentence structure should be revised and corrected throughout the manuscript.

Reviewer 2 Report

Foods

foods-2270927

Effect of Marinade Based Prepared from Valorised Food Industry By-products on Quality and Biosafety Parameters of Lamb Meat

Dear Editor,

The article deals with the development of marinade formulas based on dairy, berries and fruit industry by-products and the application of them to lamb meat treatment to improve meat safety and quality characteristics. The topic is good. However, it needs major revision. Discussion sections of the ms should be improved. My specific comments and questions;

-       Line 23: combned or combined?

-       Line 100: Please give more information about the animal such as type, sex, age, rigor situation etc.?

-       Much more information about the analyses done in the current study should have given!

-       Line 221: prepeard?

-       Lines 223 and 235: What could be the possible reasons?

-       Table 3 and 4: What does “not detected” in microbiological analyses mean?

-       Line 408: biding?

-       What about method validation parameters for biogenic amine analysis?

-       Please give the major fatty acids in SFA, MUFA, and PUFA in Table 9.

Round 2

Reviewer 1 Report

I appreciate the efforts of the authors to revise the manuscript considering all the comments and suggestions of the reviewer. The manuscript is sufficiently revised and may be accepted in its present form for the publication in Foods.

Author Response

The authors are thankful for the valuable comments.

Reviewer 2 Report

Dear Editor,

The authors have revised and improved their manuscript according to the reviewers' comments and suggestions. Therefore, the manuscript can be accepted and published in its current form.

Best regards,

Author Response

(The authors gave the same response as above.)
